# Train a snake with reinforcement learning algorithms

**Ruikang Zhang**

Department of Information Engineering

The Chinese University of Hong Kong

Shatin, Hong Kong

`1155150526@link.cuhk.edu.hk`

**Ruikai Cai**

Department of Information Engineering

The Chinese University of Hong Kong

Shatin, Hong Kong

`cr020@ie.cuhk.edu.hk`

## Abstract

Snake is a typical video game where the player maneuvers a line which grows in length, with the line itself being a primary obstacle.Our goal is to train a more efficient agent than human for Snake.To approoach this goal, firstly we use pygame achieve a simple Snake game as the environment.Because Snake game have so many states that it is impossible to use tabular method represent them and human play Snake and learn it by geting the image of it. We believe that Double-DQN which is a classical algorithm of RL is suitable for our agent training.Besides,Snake is also similar with a walking or climbing process.We note that PPO has good performance for resolving such problem by restricting the variance during training.That is why we want to compare Double-DQN with PPO in our environment.Then we use Double-DQN and PPO to train our agent.At present, we get the results in current environment.We find that Double-DQN is more stable but the peak of PPO is better.

## 1 Introduction

We have embbed a presentation video on the bottom of this report.This report need to be opened by a pdf reader with video playing function.If you do not have it,you also can find this video by this link: `https://drive.google.com/file/d/1pepbeHGFfgs0bS3ODmP0GkRjPYDXaLCi/view?usp=sharing`

## 1.1 Reinforment Learning

When we are children,what we do firstly is to try moving hands or crying and to expect getting responses.These responses will help us know about the environment where we are,the logical of different events,the result of an unique action and how to get what we want. So the growth of a child is also a process in which we interact with the outside environment.Learning from interaction is the foundament of mostly intellectual theories.Reinforcement learning is a typical learning from interaction,which is considered as the most possibel way to achieve strong artificial intelligence(6).The goal of it is to learn what should an agent do to maximize the reward.Without specifying how the task is to be achieved,it program agents by rewards and punishment.

## 1.2 Deep Reinforcement Learning

Deep reinforcement learning which incorporates both the advantages of the perception of deep learning and the decision making of reinforcement learning is able to output control signal directly based on input images(1). This mechanism makes the artificial intelligence much close to human thinking modes. Deep reinforcement learning (2)has achieved remarkable success in terms of theory and application since it is proposed. 'AlphaGo', a computer Go deve-loped by Google DeepMind, based on deep reinforcement learning, beat the world's top Go player Lee Sedol 4:1 in March 2016.Since then, more and more researchers focused on DRL, different DRL algorithms occured.

## 1.3 Snake

Snake is the common name for a video game concept where the player maneuvers a line which grows in length, with the line itself being a primary obstacle. The concept originated in the 1976 arcade game Blockade, and the ease of implementing Snake has led to hundreds of versions (some of which have the word snake or worm in the title) for many platforms. The player controls a dot, square, or object on a bordered plane. As it moves forward, it leaves a trail behind, resembling a moving snake. In some games, the end of the trail is in a fixed position, so the snake continually gets longer as it moves. In another common scheme, the snake has a specific length, so there is a moving tail a fixed number of units away from the head. The player loses when the snake runs into the screen border, a trail or other obstacle, or itself.(wiki)

## 2 Related Work

## 2.1 Deep Reinforcemrnt Learnig

In 2013, DeepMind used neural network in Atari games to help training agnets which finally perfoming beyond the level of human players on multiple games (3). It ap-

proximated value functions and policies through Policy Gradient and Deep Neural Network.Using of DNN approximationto avoid some problems from using table store data such as the large storage sequence space of tables and slow queries.Logically,the combination of DNN and RL has become a novel direction for the development of reinforcement learning.At the same time, the Actor-Critic learning method is used to cleverly realize the method of separating the value function and the strategy for learning, so that the update of the value function is guided when the strategy reaches the optimum. Since then, deep reinforcement learning has become a standard for solving problems, and the use of DNN has promoted the performance of RL in many applications. Moreover, the importance of Actor-Critic thinking has played a main role in the development of practical problems development of Q-learning.

As a landmark algorithm in the field of deep reinforcement learning, DQN (Deep Q-learning Network) has performed perfectly in Atari games (4). It approximates the value function through a neural network, uses the ideas of experience-replay and independent Target network to solve two independent distribution problems, and lays the foundation for the development of algorithms such as deep deterministic policy.

In addition, in games such as MuJoCo, since the joints of the robot are controlled by continuous variables, ordinary DQN is difficult to learn due to large-scale storage.David silver proposed the determinstic policy to solve the convergence problem.This policy have successfully played an important role in continuity control. Since then, deep deterministic policy has been further developed. and distributed deep deterministic spolicy has further improved efficiency, but deterministic policy only ensures that problems will be converged as soon as possible , Cannot ensure the optimal.It may enter the local optimal. So John Schulman of the University of Berkeley proposed a policy optimization method, which provides a monotonous policy improvement method, and the efficiency is further improved. However, the big guys are not satisfied. OpenAI and DeepMind proposed the PPO and DPPO algorithms at almost the same time. The PPO's idea (5) is to limit the update range of the new policy, not to be dizzy, to avoid "overfitting"and to adopt single-thread, while DeepMind's DPPO uses multi-thread solutions.

## 2.2  Snake

Snake is a typical Markov Decison Process which is suitable to be learned by RL.During these years,people have implemented many classical algorithms to play Snake.And with the development of DRL, some typical DRL algorithms have been used to make agents perform better in Snake.Such as DQN and DDQN.

# 3 Problem Formulation

## 3.1 Problem Definition

In a 2D environment, there are three main components: snake,boundary and fruits,corresponding to green squares,black line and white circles,respectively. In this problem, our goal is to train an virtual agent which will make decisions for the snake to decide which action to take for eating fruit and to avoid touching its own body or the boundary.

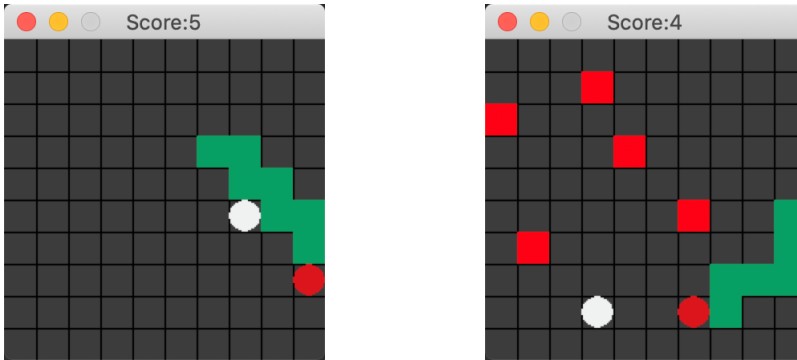

Figure 1: With and without obstacles

## 3.2 Environment and Agent

We develop our own snake environment based on pygame. Our environment includes different setups. For example we can control the scale of the environment, also we can control whether the snake can go cross the boundary or whether there are obstacles in the environment.

The environment consists of a series of squares(10*10). There are three components located in these squares: Snake,boundary and Fruit.

1.Snake is the agent in this environment. The triangle is its head, and the head controls the direction of movement, up, down, left, and right. Snke's body has three blocks at beginning.Noted, snake cannot turn back because it will be considered as touching its body. Snake once touches its own body or boundary, it will die. The goal of the snake is to eat more fruits to grow stronger and not touch other things to avoid death.

2.Boundary limits the size of the environment. At the same time, in our Double-DQN experiment, the snake will not allowed to come across the boundary once touching boundary.

3.Fruits mean reward for the snake. Environment includes only one kind of fruits: the white fruit. The white fruit appears and stays in fixed location until it is eaten by a snake and then environment will reproduce it in another place.If snake succesfully eat

a white fruit,the score it get will add one. After eating the fruit, the small snake will gain growth and increase the length of its body.

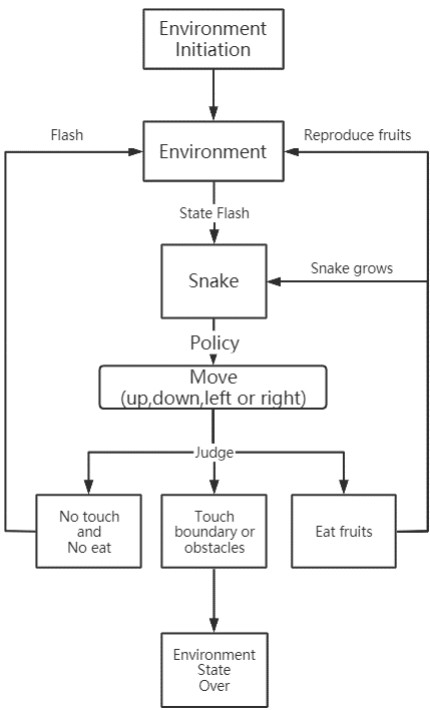

Figure 2: Environment and Agent

## 3.3 Reward

1.Firstly we define the distance between snake and fruit as follow:

$$dis = \sqrt{(x_1 - x_0)^2 + (y_1 - y_0)^2}$$

$(x_0, y_0)$ is the positon of the fruit.

$(x_1, y_1)$ is the postion of snake head

2. Then we can formulate the reward for the snake as following table:

| State | Reward |
|-------|--------|
| take a step and nothing happen | (1/max(1.0,dis)*1 |
| take a step and game over | -1 |
| take a step and eat a fruit | +1 |

## 4 Experiments

### 4.1 Double-DQN Algorithm

We select Double-DQN algorithm to get the more efficient agent.

### 4.1.1 Approximator

Double-DQN use non-linear value function approximation,in our experiment,we choose CNN as the action-value function approximation.

Input state s is raw pixels from replay memory.Output of $Q(s,a)$ is the action-value for every action.

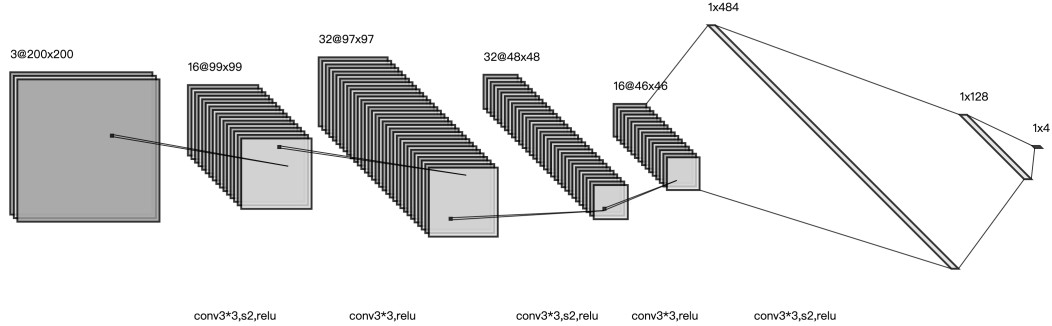

Figure 3: CNN

### 4.1.2 Experience replay and fixed Q-targets

Besides,due to correlations between samples and noe-stationary targets, Double-DQN used experience replay and fixed Q-targets to address these two issues.

1.Experience replay:

Store transition $(s_t, a_t, r_{t+1}, s_{t+1})$ in replay memory $D$

$s_t$ is the state of step t

$a_t$ is the chosen action on step t

$r_{t+1}$ is the reward after$(s_t, a_t)$

$s_{t+1}$ is the state of next step t+1

Sample random mini-batch of transitions$(s, a, r, s')$ from $D$

2.Fixed Q-targets

Let a different set of parameter $W^-$ be the set of weights used in the target, and $W$ be the weights that are being updated

Compute Q-learning targets with respect to old,fixed parameter $W^-$

$$r + \gamma * max_{a'} \hat{Q}(s', a', w^-)$$

### 4.1.3 Optimizer

Optimizes MSE between Q-network and Q-learning targets using stochastic gradient descent

$$\Delta w = \alpha * (r + \gamma * max_{a'} \hat{Q}(s', a', w^-) - Q(s, a, w)) * \nabla w \hat{Q}(s, a, w)$$

$\alpha$ is the learning rate

$r$ is the reward

$\gamma$ is the discount

$s'$ is the next state

$a'$ is the next action

$w$ is the weight of Q-behavior

$w^-$ is the weight of Q-target

$\hat{Q}(s', a', w^-)$ is the Q-target function

$Q(s, a, w)$ is the Q-behavior function

### 4.1.4   Double-DQN Training

We trained our agent and store every better evaluate result during the training process. We trained and got our final Double-DQN agent from about 140 thousand steps. This took about 80 hours in Mac book CPU.

We repeatedly use the latest agent to run Snake until death 30 times and compute the average score. The train result shows as following figure.The agent can get 10.4 fruit each game at average

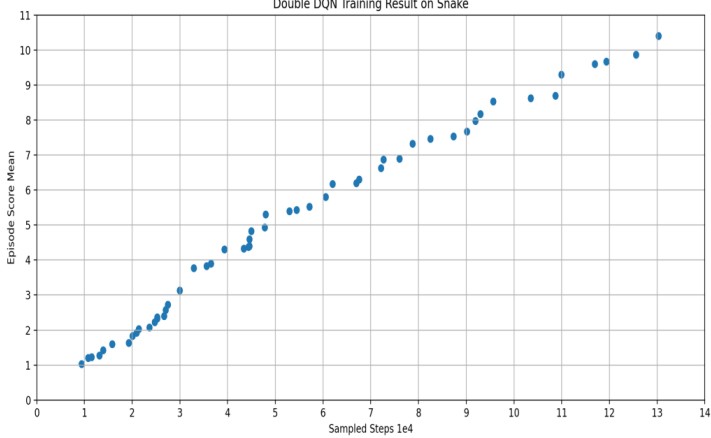

Figure 4: Progress of DDQN

The agent can eat 15 fruits at most.

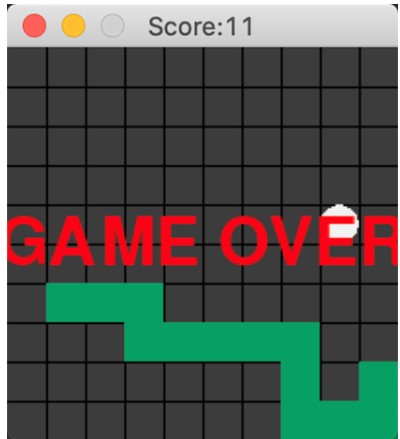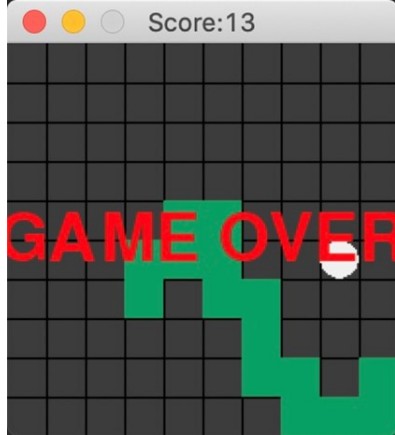

Figure 5: Result of DDQN

## 4.2 PPO

We use PPO as the comparison method to train our agent.

### 4.2.1 PPO Algorithm

We choose the PPO algorithm as the agent. PPO collects a small batch of experiences interacting with the environment and uses that batch to update its decision-making policy. Once the policy is updated with this batch, the experiences are thrown away and a newer batch is collected with the newly updated policy. The key contribution of PPO is ensuring that a new update of the policy does not change it too much from the previous policy. This lead to less variance in training at the cost of some bias, but ensures smoother training and also makes sure the agents does not go down an unrecoverable path of taking senseless actions.

---

**Algorithm 1** PPO with Clipped Objective

---

**Input:** initial policy parameters $\theta_0$, clipping threshold $\epsilon$

1: some description
2: **for** $k = 0, 1, 2, 3...$ **do**
3:      Getting partial trajectories of Snake $D_K$ on policy $\pi_k = \pi(\theta_k)$
4:      Estimate advantages $\widehat{A}_t^{\pi_k}$ using advantage estimation algorithm
5:      Compute policy update

$$\theta_{k+1} = (L_{\theta_k}^{CLIP}(\theta))$$

6:      by taking K steps of minibatch SGD (via Adam)

---

Meanwhile, we use CNN as the actor-critic networks. CNN is a neural network designed for image recognition and it perform well in the image recognition. The structue of CNN is similar with CNN we use in Double-DQN.The only difference between them is we use CNN in PPO as the approximater of both value and policy.

### 4.2.2 PPO Training

We use PPO to train our agent and make an estimate every time we learn 50 times.This snake will walk about 1000 steps in a learning process. After learning 900 times,the score dose not go further.The average score obtained is shown in the figure. After 900 times of learning, the final result is 7.89.This means the agent can get 7.89 fruits at average.

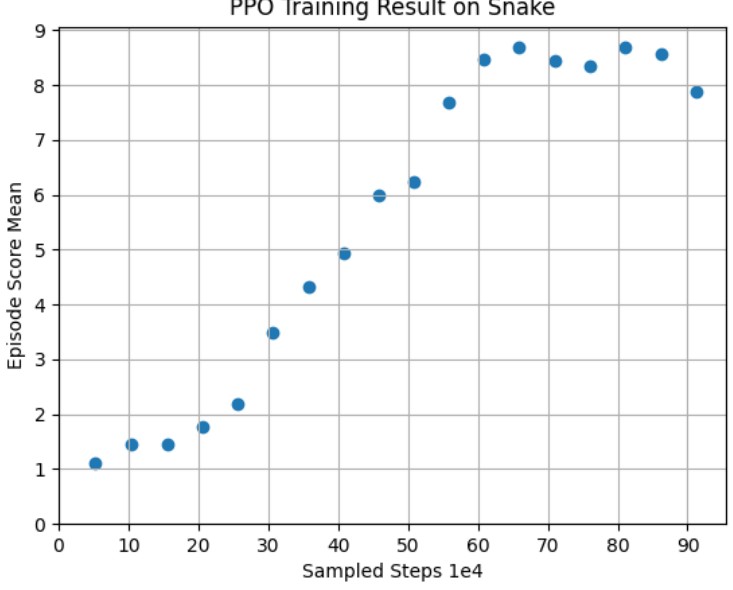

图 6: Progress of PPO

The agent can get 18 fruits at most.

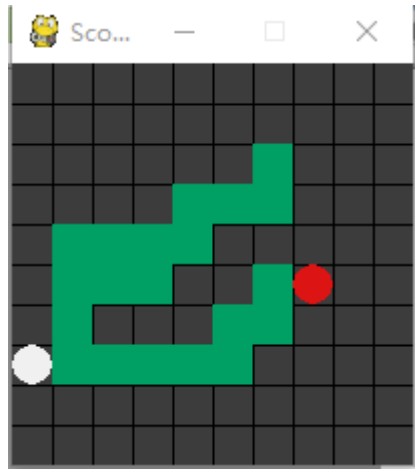

图 7: Result of PPO

## 5 Conclusion

In our project,we create an environment of Snake game.Then by training our agent using different deep learning algorithm in the same environment,we get the two results of Double-DQN and PPO.We find that the performance of Double-DQN's agent is more stable.The result of PPO is not satisfied.The possible reasons are restriction of the environment and the flaw of network.We plan to explore more about PPO in future research.

| Algorithm | Mean score | Best Score | steps |
|-----------|-----------|-----------|-------|
| Double-DQN | 10.3 | 13 | 140k |
| PPO | 7.89 | 18 | 900k |

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
