# OpenReview forum: "Train a snake with reinforcement learning algorithms"
_CUHK.edu.hk/2021/Course/IERG5350_

### Official Review · AnonReviewer1 · 2020-12-17
**An implementation of resolving the Snake problem using Double-DQN and PPO**

**Rating:** 4
**Confidence:** 5

**Review:**

Summary:
This work tries to resolve the Snake problem in a self-written Snake environment using Double-DQN and PPO. The results show that the agent trained by Double-DQN is more stable but achieves a lower threshold reward than the agent trained by PPO.

Comment:
1. The authors spend a lot of words to give an introduction to the background of reinforcement learning, which I think is unnecessary as we the reviewers are teachers and students of this course, who should be familiar with this information.
2. As the Snake problem is popular in reinforcement learning and the authors should state some related implementations, or implement some of the existing works in their self-written Snake environment and show the novelty of the author's work.
3. The authors implemented a $10 \times 10$ Snake environment in Pygame, which is not novel as I can simply find an implementation [1] in Google which has a larger map size than this one. Also, the abovementioned implementation has achieved a much better result than this work.
4. In Figure 1, the authors show an environment filled with obstacles (the red blocks; which I do not know if they are randomly generated or not) and an environment without the obstacles. Hence, there should be two sets of results but I can only see of one environment (I assume it is the result of the environment without obstacles from figure 5 and 图7).
5. I think implementing an environment filled with randomly generated obstacles and successfully train agents to resolve this problem will be a novelty, and is expected in this work.
6. No codes of the experiments conducted in this work can be publicly accessed.

Reference:
[1] https://towardsdatascience.com/snake-played-by-a-deep-reinforcement-learning-agent-53f2c4331d36

---

### Official Review · AnonReviewer2 · 2020-12-17
**Interesting exploration on playing Snake with RL, but the paper still needs some major revision.**

**Rating:** 5
**Confidence:** 4

**Review:**

## Summary:
Authors of this paper try to play the Snake game with reinforcement learning. They implement their own game environment with several settings (e.g., with or without obstacles) and use DQN (and PPO) to train it. The trained agent can achieve over 10 scores on average in the game.

## Strength:
* They develop their own Snake game for RL, with different settings.
* The trained agent can play the game with reasonable performance.
* The authors tried two algorithms on their environment, PPO and DDQN, and compared their performance.

## Weakness:
* Lack of review and comparison with existing works, especially the existing RL-based solutions for the Snake game.
* Needs more detailed evaluations: how many episodes for each evaluation/estimate, it's unclear whether high _best score_ is caused by small sample size and large variance. Also, there's no hyperparameters tuning/reasoning.
* The writing and formatting need improvement.
* This paper would be more interesting if you do some experiments with the _Snake with obstacles_ setting, even if preliminary.

## Comments:
1. It's better if you can add some citation in 2.2 to elaborate related works.
2. How's your trained agent comparing with expert human players? It is unclear that at what level your trained agent's performance is.
2. From the curve of DDQN, it seems the training hasn't converged yet. I understand that you may not have enough time to train it. Have you considered speed up your training process first? For instance, try to use internal data instead of pixel input. After all, you already use internal knowledge when calculating the distance reward.
3. The formatting of the paper looks a little bit messy: e.g., missing space after many punctuations, tables without titles, formula with a long list of notation description.
4. The paper needs a grammar pass. Please proof read it to correct typos and grammar errors.

---

### Official Review · AnonReviewer3 · 2020-12-17
**This article uses Double-DQN and PPO to train an agent in the Snake game and compare the results.**

**Rating:** 6
**Confidence:** 4

**Review:**

General:
1. Significance: This article uses Double-DQN and PPO to train an agent in the Snake game and compare the results. It meets the basic requirement of this course project.
2. Novelty: There is no innovation or improvement,  although it explains the basic process of algorithms.
3. Technical quality: It does not mention any hyper-parameter setting for Double-DQN and PPO or implement different parameters to do further experiments.
4. Clarity: This article is not concise enough.  There are too much unnecessary content in the background part. The authors should add a blank space after each punctuation and pay attention to grammar errors.

Specific:
1. Pros:  Double-DQN and PPO are implemented in the game.
2. Cons: a. No improvements or optimazations in Double-DQN and PPO. b. The experiment and conclusion are quite simple.

---

### Official Review · AnonReviewer4 · 2020-12-19
**The topic of this project is a bit simple and more explorations are needed.**

**Rating:** 4
**Confidence:** 5

**Review:**

Report summary:

This report describes a Snake game played by reinforcement learning. Two training algorithms, i.e.,g Double-DQN and PPO are adopted. Experiments demonstrated that Double-DQN is more stable but PPO achieves a better peak performance.


Major issues:

1. The topic of this project is too simple and there are already many codes available online for reference.
2. Although the topic is simple, I think it is ok to implement one by yourself. However, more explorations should be conducted. For example:
* More interesting settings for the environment, e.g., random obstacle appearing in the environment.
* Try different metrics for calculating the distance between the fruit and the head of the snake, e.g., city block.
* The sensitivity of different algorithms to the size of the environment in terms of performance, efficiency, etc.
3. A thorough proofreading is needed! There are numerous grammatical mistakes and typos in the report. The writing also needs improvement. Some suggestions are listed below:

Writing:
* All English punctuations should be followed with a space
* Spell out an abbreviation at its first occurrence

Grammatical mistakes and typos:
* Line 4 of Abstract: we use pygame achieve -> we use pygame to achieve
* Line 7 of Section 1.1: possibel -> possible
* The last line of Section 1.1: it program -> it programs
* etc.


Minor issues:
1. It is said that the head of the snake is represented with a triangle, but I do not see it.
2. I do not see any explanation for the red circle and red rectangle.
3. The formatting of the report should be improved.